# Physiological Changes of Arabica Coffee under Different Intensities and Durations of Water Stress in the Brazilian Cerrado

**DOI:** 10.3390/plants11172198

**Published:** 2022-08-25

**Authors:** Patrícia Carvalho da Silva, Walter Quadros Ribeiro Junior, Maria Lucrecia Gerosa Ramos, Omar Cruz Rocha, Adriano Delly Veiga, Nathalia Henriques Silva, Lemerson de Oliveira Brasileiro, Charles Cardoso Santana, Guilherme Filgueiras Soares, Juaci Vitória Malaquias, Christina Cleo Vinson

**Affiliations:** 1Faculdade de Agronomia e Medicina Veterinária, Universidade de Brasília, Brasília 70910970, DF, Brazil; 2Embrapa Cerrados, Empresa Brasileira de Pesquisa Agropecuária, Planaltina 73310970, DF, Brazil; 3Embrapa Café, Empresa Brasileira de Pesquisa Agropecuária, Brasília 70770901, DF, Brazil; 4Departamento de Agronomia, Universidade Federal de Viçosa, Viçosa 36570900, MG, Brazil

**Keywords:** *Coffea arabica*, genotypes, coffee physiology, drought

## Abstract

Coffee farmers have faced problems due to drought periods, with irrigation being necessary. In this sense, this study aimed to evaluate the responses to different levels and durations of water deficit in arabica coffee genotypes in the Cerrado region. The experiment consisted of three *Coffea arabica* genotypes and five water regimes: full irrigation (FI 100 and FI 50—full irrigation with 100% and 50% replacement of evapotranspiration, respectively), water deficit (WD 100 and WD 50—water deficit from June to September, with 100% and 50% replacement of evapotranspiration, respectively) and rainfed (without irrigation). The variables evaluated were gas exchange, relative water content (RWC) and productivity. The results showed that during stress, plants under the FI water regime showed higher gas exchange and RWC, differently from what occurred in the WD and rainfed treatments; however, after irrigation, coffee plants under WDs regained their photosynthetic potential. Rainfed and WD 50 plants had more than 50% reduction in RWC compared to FIs. The Iapar 59 cultivar was the most productive genotype and the E237 the lowest. Most importantly, under rainfed conditions, the plants showed lower physiological and productive potential, indicating the importance of irrigation in *Coffea arabica* in the Brazilian Cerrado.

## 1. Introduction

Coffee (*Coffea arabica*), a tropical perennial tree species, is an agricultural commodity, traded globally and responsible for the livelihoods of 25 million small farmers, constituting the economic basis of tropical developing countries [1]. It is one of the most important agricultural products on the international market, cultivated at different latitudes worldwide. However, its distribution depends on many climatic factors, such as location, soil types, shading and management practices [2].

In Brazil, *C. arabica* was initially cultivated and developed in regions without water deficits during periods of harvest. With the expansion of agriculture in the Cerrado areas and the control of water deficits with irrigation, this crop expanded to areas with higher temperatures, such as the Triângulo Mineiro region and the northeast region of Minas Gerais, west of Bahia, state of Goiás and Federal District [3]. Nowadays, Brazil is the largest *C. arabica* and *C. robusta* producer in the world.

However, studies predict that, in Brazil, climate change will drastically reduce the suitable areas for growing this crop due to an increase in temperature and a decrease in rainfall [4,5]. Additionally, the advent of plants for energy production, as well as sugar cane, will displace other crops to marginal and limiting areas.

Drought is the main environmental stressor that affects coffee production, causing a reduction in productivity of up to 80% in very dry years and/or in marginal regions without irrigation [6]. It is shown that the technique of controlled water stress in irrigated coffee trees in the flowering period standardizes flowering and generates greater production of coffee in cherry, and thus generates a potential increase to produce specialty coffees, which have a better price in the market [7]. Therefore, identifying and understanding the physiological mechanisms that protect the plant during water stress and accelerate recovery from stress associated with the use of this information in the genotype selection process are extremely important, as the ability of plants to recover physiological functioning after the onset of irrigation represents the key point of plant resilience to drought.

Water stress has multiple effects on plant physiology and development that can result in reduced productivity. Water absorption and generation of turgor pressure is necessary for cell expansion and division; hence, water deficit directly reduces organ growth as leaf water potential and relative water content decrease. Plants typically respond to water stress by closing their stomata to reduce transpiration. CO_2_ entry into the leaves is therefore reduced and consequently photosynthesis as well, resulting in reductions in carbon fixation and production of the photosynthate needed for both vegetative and reproductive growth [8]. Reduced transpiration will also tend to increase leaf temperature, which may result in damage to the photosynthetic apparatus, and oxidative stress may develop due to the production of reactive oxygen species. Xylem water transport may also be impacted as low water potentials result in cavitation and difficulty in transporting water from the roots to leaves [9]. Finally, specific developmental effects of water deficit may occur, including the abortion of flowers and embryos, alterations in the timing of flowering and loss of leaves.

Although it results in decreased photosynthesis, stomatal closure represents an important mechanism in plant water deficit tolerance. Genotypes that show greater or more rapid stomatal closure in response to low soil water availability, which can be evaluated using parameters such as stomatal conductance (*gs*), may show improved water use efficiency and fewer symptoms of stress with maintenance of turgor [10]. Indeed, stomatal control appears to be an important factor in determining tolerance to drought stress in coffee [11]. At the cellular level, the accumulation of osmolytes including sugars and inorganic ions can help to maintain water status and turgor, as well as protecting proteins, nucleic acids and membranes. Synthesis of antioxidant proteins and metabolites can reduce the severity of oxidative stress. Morphological differences are also important, with smaller leaves typically associated with improved cooling and better stress tolerance and a more extensive root system permitting greater water absorption from the soil [11].

In arabica coffee, a key component of the differential adaptation between cultivars appears to be the stomatal control of water loss and the efficiency of soil water extraction [11]. However, physiological mechanisms that increase water use efficiency (WUE) may conflict with those related to growth and productivity [12]. Therefore, the identification of cultivars combining satisfactory growth and yield and high WUE is of great relevance for coffee cultivation in drought-prone areas, in environmental conditions that were previously unfavorable for crop cultivation and for the sustainability of the production system. 

For this, evaluations after crosses in breeding programs, with many genotypes, under field conditions are necessary. Some of these evaluations are destructive, such as plant biomass, and demand more resources, space and time as they require the transportation of part of the plants from the field to the laboratory [13]. Here, we adopted a different approach using a non-destructive method, using an infrared gas analyzer—IRGA—and an efficient methodology to characterize genotypes in the field by physiological attributes when exposed to different water regimes and in recovery from water stress. In this context, we hypothesized that water stress affects physiological plant traits, and coffee genotypes differ in relation to responses to water addition. Thus, the objective of this work was to evaluate the effects of different levels and durations of water deficit in arabica coffee genotypes under field conditions, characterize genotypes and search for mechanisms of drought tolerance and irrigation responsiveness for coffee cultivation in the Brazilian Cerrado.

## 2. Results

### 2.1. Photosynthetic Analysis

#### 2.1.1. Daily Curve

The gas exchange variables showed similar trends throughout the day. However, among the genotypes, a variation was observed in relation to the highest peak of gas exchange during the day (Appendix A). Iapar 59 had the highest peaks of stomatal conductance (gs), transpiration (*E*) and photosynthesis (*A*) between 10:00 am and 1:00 pm. In contrast, the highest peak of Catuaí 62 occurred from 7 am to 9 am, while E237 showed higher stomatal conductance and photosynthesis than the other genotypes, and its highest peak was between 8:00 am and 9:00 am, and its lowest peak at 11:00 am; from this time onwards, there was a significant drop in gas exchange (Appendix A).

#### 2.1.2. Gas Exchange—Genotypes x Recovery

The gas exchange data of the three genotypes during stress and post-irrigation in the treatments with water deficit, in the two years, are presented in Figure 1, Figure 2 and Figure 3. In the FI treatments (FI 100 and FI 50), there was no recovery irrigation due to irrigation taking place all year round in 2019 and 2020. Similarly, in the rainfed treatment, the evaluations represent the physiology of coffee plants throughout the dry period in the Cerrado, with the possibility of recovery only with rain (Figure 1, Figure 2 and Figure 3I,J).

The stomatal conductance and transpiration of the three genotypes during the evaluations in FI 100 and FI 50 showed similar results. Although the treatments with full irrigation, FI 100 (Figure 1A) and FI 50 (Figure 1C), had the same trends, in 2019, the values were much lower in the treatment that received only half irrigation (FI 50), and this was not observed in 2020.

Physiological recovery in the WD 100 and WD 50 treatments was slow at the beginning of the return of irrigation, indicating that the soil had low moisture (Appendix A) and the plants had low water content in the leaves (Figure 1, Figure 2 and Figure 3E–H). However, from Assessment 4 onwards, gas exchange started to cause a more prominent increase in all genotypes (Figure 1, Figure 2 and Figure 3E–H).

#### 2.1.3. Gas Exchange—Genotypes x Water Regimes

We also evaluated the gas exchange of the three arabica coffee genotypes using different water regimes, during the water stress period and after the return of irrigation in treatments with water deficits (Figure 4, Figure 5 and Figure 6). Due to the water supply in FI 100 and FI50, the genotypes presented higher stomatal conductance and transpiration in the FIs, different from what occurred in WD and rainfed treatments (Figure 4 and Figure 5A,C). On the other hand, there was a significant decrease in transpiration in the genotypes in rainfed and WD due to reduced water availability and consequent stomatal closure (Figure 4 and Figure 5A,C).

The photosynthetic rate of the coffee genotypes, in general, was higher in the FI treatments (Figure 6A,C). E237 had lower results in rainfed conditions in 2019 (0,6 µmol m^−2^ s^−1^) and higher in 2020 (1.14 µmol m^−2^ s^−1^) (Figure 7A,C). In the evaluation performed after the irrigation, the plants subjected to water stress recovered their photosynthetic potential, with an increase of approximately 60% in the photosynthetic rate of the genotypes, compared to the stress assessment in 2019. In some genotypes, the values were similar, and no significant difference was obtained (Figure 4, Figure 5 and Figure 6B,D).

#### 2.1.4. Relative Water Content

The relative water content (RWC) was higher in leaves with full irrigation (>70%, with the exception of E237) treatments, in all genotypes, due to the greater availability of water in the plants in these water regimes (Table 1). Rainfed and WD 50 had a more than 50% reduction in RWC compared to FIs. Except for E237, no difference was observed in the relative water content of the genotypes in FI 100 and FI 50. On the other hand, in treatments with water deficit and in rainfed conditions, RWC was below 50%. As observed for gas exchange, the 2020 RWC data in the deficit and rainfed treatments were higher than those in 2019 (Table 2).

In the FI 100 treatment in 2020, the E237 genotype presented lower RWC (61.72%) in the leaves than the other genotypes (Table 1). On the other hand, in 2019, in the rainfed treatment, E237 and Iapar 59 had higher RWC (38%) than Catuaí 62 (32%), demonstrating that in the rainfed treatment, E237 maintains greater hydration and retains more water than when it is subjected to treatments with irrigation.

There was no significant interaction between the genotypes and water regimes during the evaluation after the return of irrigation; therefore, they are discussed separately. When evaluating the genotypes, we observed that E237 presented more than 80% water content in the plants after irrigation, different from what occurred in the evaluation of plants under stress (Table 1), indicating the rapid recovery of the RWC of this genotype (Table 2). The effects of water regimes on the RWC of plants recovered by irrigation in 2019 show that treatments with a water deficit in some cases maintained the RWC of the leaves. In 2020, due to rain before the last evaluation, the recovery data showed little difference between treatments (Table 3), with the lowest RWC in the rainfed treatment.

### 2.2. Productivity

The highest coffee productivity was observed in the FI and WD 100 treatments, depending on the genotype and year (Figure 8). The higher water deficit in the WD 50 and rainfed treatments was reflected in the plants’ productivity, and these treatments yielded lower production in all studied genotypes. The treatment with adequate water (WD 100) significantly reduced productivity in 2020 in the Catuaí 62 and Iapar 59 genotypes. In E237, there was no significant difference in productivity in the FI and WD 100 treatments, which would be the ideal situation. In 2019, all genotypes showed lower productivity. E237 and Catuaí 62 were more productive under the FI 50 and WD 100 treatments, whereas Iapar 59 showed higher productivity under the FI and WD 100 treatments.

## 3. Discussion

### 3.1. Gas Exchange 

Coffee is a shade-adapted crop and has very low saturation irradiance for leaf photosynthesis compared to other crops. However, these values depend on the genotype studied, as shown in Appendix A. Photosynthesis occurred in the range of 4—11 µmol m^−2^ s^−1^ in conditions of atmospheric CO_2_ concentrations and saturating light. According to the study by Martins et al. [14], photosynthesis in coffee is mainly limited by stomatal factors, followed by limitations associated with mesophyll and biochemical restrictions [15,16]. Our data show that the ideal period to carry out gas exchange evaluations in coffee is between 7 and 11 am, depending on the genotype, to obtain a better evaluation of the plant’s physiological activity, because, at warmer times of the day, there is a reduction in stomatal conductance, which directly results in a decrease in photosynthesis.

In the daily photosynthesis curve, photosynthesis performed in the afternoon for the genotypes Iapar 59 and Catuaí 62 demanded a high transpiration rate from the plants due to greater stomatal opening (Appendix A). On the other hand, the E237 genotype showed lower transpiration during the hottest period of the day due to strong stomatal sensitivity and an increase in the vapor pressure deficit, which largely restricts the CO_2_ influx into the leaves, limiting photosynthesis, reducing the loss of water by transpiration and avoiding plant desiccation [11]. This genotype has Ethiopia as its center of origin and has been used in controlled crosses as a gene source by Embrapa Cerrados. Knowledge of the dynamics of gas exchange among genotypes under different environmental conditions is useful in choosing parents for breeding programs for a given region.

Gas exchange parameters such as net photosynthesis, transpiration and stomatal conductance are sensitive indicators of water deficit in plants. They are useful to evaluate genotypes adapted to environments with limited water availability [17,18]. Furthermore, stomatal closure is known to be an important component of the drought response of coffee [11]. Therefore, we evaluated physiological characteristics in coffee genotypes under water deficit conditions and verified that the slow recovery of stomatal conductance in plants under water deficit indicates the sensitivity of the stomata to water availability (Figure 1). Similarly, in vines, researchers observed the slow recovery of stomatal conductance after a prolonged water deficit, and the delay was attributed to the foliar accumulation of ABA [19]. Other parameters may also be valuable, however, and an analysis of eleven tree species for use in reforestation in drought-prone areas indicated high values of leaf mass per area and midday water potential, together with low antioxidant levels and 13C discrimination, as being associated with drought resistance [18].

Our results suggest that the reduction in photosynthesis when plants are under stress is related to stomatal limitations, due to the stomatal conductance of coffee being lower than that of other tropical plants, with values lower than 0.02 mol m^−2^ s^−1^, which is in agreement with the results of Martins et al. [14]. Furthermore, since CO_2_ and water molecules share the same pathway at both the leaf and cell levels [20], the reduction in stomatal conductance will restrict the photosynthetic efficiency of plants, as it is considered the main cause of regulation photosynthesis during water stress [21].

The suspension of irrigation in WD treatments reduced transpiration, with values below 0.2 mmol m^−2^ s^−1^ (Figure 5A,C). Although reducing water loss through transpiration is important for the temporary maintenance of the water potential in the leaf [22], decreased transpiration through stomatal closure results in less biomass accumulation due to decreased carbon assimilation, reflecting directly in photosynthetic activity and crop productivity, as observed in plants under the WD 50 and rainfed treatments (Figure 6 and Figure 7). With this, we realized that the characteristics that increase the efficiency of water use may have conflicted with those related to crop productivity and resulted in low productivity in treatments with greater water deficits.

After the return of irrigation, in the treatments with WD 100 and WD 50, the plants recovered their photosynthetic potential (Figure 4, Figure 5 and Figure 6). Thus, at the time of full flowering (September/October) of the coffee tree, they already had the ideal water status for the full development of the crop (Table 1). Our data are consistent with previous studies showing that the time required for recovery depends on the severity of water stress, as it determines the extent to which physiological functions are impaired [23,24].

The photosynthetic parameters show that genotype Iapar 59 develops well in water conditions, as observed in Figure 6, whereas Iapar 59 presents the highest photosynthesis in the WD treatment, equal to E237 in some estimates. The E237 genotype was introduced from Ethiopia, and the study of Carvalho et al. [25] shows that Iapar 59 and Ethiopian introductions are drought-tolerant. Likewise, the Geisha coffee plant, introduced in Ethiopia, showed drought tolerance under field conditions [26], and Iapar 59 was considered tolerant to water deficit after obtaining greater expression of the M6PR gene, responsible for the expression of biotic and abiotic stress responses [27]. Thus, Iapar 59 can be cultivated in irrigated areas or areas prone to water deficit, and E237 can be used in crosses with other materials in breeding programs. From an ecophysiological point of view, drought-tolerant coffee cultivars are able to sustain a better water status during periods of long-term drought, which has been attributed to a combination of deep rooting and adequate stomatal control [28,29].

### 3.2. Relative Water Content

The relative water content (RWC) is a good reference for the water conditions of the plant as it represents the balance between water supply and transpiration [30]. The FI treatments had similar RWC, indicating efficiency in the use of water by the plants under irrigation with 50% because, depending on the year, there was no significant difference in the productivity of FI 50 and FI 100 (Table 1 and Figure 7).

According to Barrs and Weatherley [31], RWC values of around 98% indicate turgid leaves, and those between 30 and 40% reveal plants with water scarcity. In this way, rainfed and WD 50 plants in 2019 had water shortages, with RWC around 31%, compromising the photosynthetic potential of plants under this treatment, as seen in Figure 8. Thus, our data indicate that these treatments are not suitable for coffee production in the Cerrado. The decrease in the photosynthetic potential of these plants, together with the decline in foliar RWC, confirmed the lower efficiency of CO_2_ assimilation under stress due to water limitation.

Under rainfed conditions, the E237 genotype had satisfactory performance. In addition, it was able to recover its water content in the leaf faster than the other genotypes (Table 1). As coffee is a perennial crop with long vegetative and reproductive cycles, the recovery is a key point in the plant’s resilience to drought. Furthermore, fast recovery is associated with efficient stomatal control that minimizes water loss through transpiration, considered a water-saving species rather than a dehydration-tolerant species [32].

### 3.3. Productivity

As it is a biennial species, there are morphological, physiological and productive differences in the plant between the studied years in this work. As photosynthetic parameters were lower in 2019, this was reflected in lower productivity. In 2019, a more accentuated influence of the plant’s biennial nature was observed in the treatment of FI 100, as the traditional management of coffee trees in the Brazilian Cerrado conditions, with irrigation throughout the year, results in a large annual variation in productivity (accentuated biennially) and non-uniformity in the maturation of the grains, due to the occurrence of multiple blooms [28]. However, a drought period, as proposed by Guerra et al. [33] and validated by Silva [7] and Veiga et al. [3], is of fundamental importance for flower uniformity and fruit maturation. 

The lower water availability in the WD 50 and rainfed treatments led to a significant decrease in production in all genotypes studied, not being indicated for coffee cultivation in the Cerrado—that is, it is necessary to subject the plants to a moderate water deficit to standardize the flowering, as proposed by Guerra et al. [33]. However, it is not recommended to reduce irrigation to 50% of the evapotranspiration. Silva et al. [7] obtained higher productivity in irrigated treatments or in plants subjected to mild water deficits, but also demonstrated that it is possible to combine uniformity in fruit maturation and productivity, if the appropriate deficit level is applied for each region.

We observed that the genotypes presented different characteristics and responses to water availability. E237 belongs to the Embrapa Cerrados germplasm bank, with an origin in Ethiopia and characteristics of large size, long internodes and lower grain yield than cultivars available on the market, as observed in all treatments (Figure 7). However, it showed faster recovery of gas exchange and RWC in treatments with water deficits (WD 100, WD 50 and rainfed). Iapar 59 had high physiological and productive potential in regimes with high water availability and water deficits.

## 4. Materials and Methods

### 4.1. Characterization of the Experimental Area

The experiment was carried out at Embrapa Cerrados—Planaltina DF (15°35’ S and 47°42’ W, at an altitude of 1007 m). According to the Köppen classification [34], the region is characterized by an Aw climate, with two well-defined seasons (dry and rainy) and average annual temperatures of 21.1 °C and rainfall of 1.345 mm. The rainy season occurs between October and April, and the dry season between May and September (Figure 8). The soil of the experimental area is classified as clayey Oxisol (Typic Haplustox) [35], with a soft undulating relief and a clayey texture.

### 4.2. Experimental Setup

The study was initiated in April 2015 in an area of 7.359.5 m^2^ (0.74 ha^−1^), divided into five experiments: six irrigated and one without irrigation (Appendix A). Each experiment received a different intensity and duration of water regime: (1) FI 100 (full irrigation with 100% replacement of evapotranspiration), (2) FI 50 (full irrigation with 50% replacement of evapotranspiration), (3) WD 100 (water deficit with the suspension of irrigation from June to September and after water stress, replacement of 100% of evapotranspiration), (4) WD 50 (water deficit with the suspension of irrigation from June to September and after water stress, replacement of 50% of evapotranspiration), (5) rainfed (no irrigation).

The genotypes used were Iapar 59 (Sachimor and canephora gene carrier), Catuaí 62 (Yellow Caturra IAC 476-11 with Mundo Novo) and E237 (*C. arabica* accession from Ethiopia), with Iapar 59 classified as having intermediate tolerance and Ethiopian access (E237) as high tolerance to water tress [25]. The seedlings were planted at 3.50 m between rows and 0.50 m between plants, with a density of 5600 plants per hectare. The plot consisted of eight plants, and three plants were used for the evaluations.

All plots received annually 400 kg ha^−1^ of N as urea and potassium sulfate distributed in four doses in February, March, September and December, with 300 kg ha^−1^ of P_2_O_5_ in two doses (2/3 were applied in September and 1/3 in December). Micronutrients were applied at a dose of 100 kg ha^−1^ in December, as FTE-BR12, with the composition of (in %) 3.2 S; 1.8 B; 0.8 Cu; 2.0 Mn; 0.1 Mo; 9.0 Zn; and 1.8 Ca.

Water stress treatments started in April 2017, when the plants were approximately two years old. Every year, the same water regime was applied in each treatment, and the water accumulation in all treatments in 2019 and 2020 is presented in Appendix A. The water regimes were applied by a linear irrigation system pulled by a self-propelled reel and calculated by the Cerrado Irrigation Monitoring program [36]. Figure 9 shows the irrigation system used in field conditions. 

The experimental design was in the form of randomized blocks with four replications. Due to the impossibility of randomizing the water regimes, each water treatment constituted an experiment with four replications of each material, constituting a group of experiments.

### 4.3. Variables Analyzed

#### 4.3.1. Physiological Variables

Gas exchange evaluations (photosynthesis, stomatal conductance, internal concentration of CO_2_ and transpiration) were performed during water stress and after the return of irrigation in all water regimes in 2019 and 2020. The evaluations took place in August (during water stress) and after the return of irrigation (from September to October). Several evaluations were performed during the recovery from water stress (six evaluations) to monitor the physiological recovery of plants after the return of irrigation in treatments with water stress and corresponded to the first, second, fourth, seventh and tenth day of recovery irrigation, respectively.

A portable open–flow gas exchange system (IRGA–LI-6400XT; LI–COR Inc., Lincoln, NE, USA) was used for gas exchange analysis. Evaluations were performed on three plants per plot, on the youngest fully expanded leaves of plagiotropic branches, in the middle third of the plants. Gas exchange was determined with a CO_2_ concentration of 400 ppm, a photosynthetic photon flux density of 1000 µmol m^−2^ s^−1^ and relative humidity between 40% and 60%. The evaluations were performed in the morning, between 8 am and 11 am; this period of evaluation was determined by a daily photosynthesis curve (Appendix A) produced during the full irrigation treatment from 6 am to 6 pm, with an interval of one hour between each evaluation.

#### 4.3.2. Soil Water Content and Relative Water Content in Leaves

At each physiological measurement, soil samples were collected at the depth of 0–20 cm and soil water content was evaluated by the following equation: (1)$$SWC\%=\frac{WW-DW}{DW}\times 100$$
where *SWC* is soil water content, *WW* is soil wet weight and *DW* is soil dry weight (Appendix A).

The relative water content in leaves was evaluated under stress and after recovery and was determined by collecting ten leaf discs of 0.8 cm in diameter for each repetition, obtained from leaves collected in the morning, between 8 and 9 am, and determined from the following equation:(2)$$RWC\%=\frac{FM-DM}{TM-DM}\times 100$$
where *FM* is the fresh mass, *TM* is the turgid mass after rehydration of the leaf discs by immersion in distilled water for 24 h in the dark and *DM* is the dry mass of the leaf discs after drying in an oven at 60 °C, according to the methodology of Barrs and Wheaterley [31].

#### 4.3.3. Productivity

Coffee berries were harvested between April and June by stripping them on a cloth, and their volume was determined. The samples were dried on a terrace, and their volume and weight were quantified. The yield was estimated with a standardized moisture content of 13% and calculated according to the number of plants per plot. Productivity was converted into bags ha^−1^ for each genotype and water regime.

All variables were assessed in two years of study, 2019 and 2020, to increase the data’s reliability due to the culture’s biennial nature.

### 4.4. Statistical Analysis

For the statistical analysis of data, water regimes and genotypes were considered sources of variation. Data were subjected to analysis of variance at 5% probability using the F test, and the means were compared using Tukey’s test. To verify the feasibility of proceeding with the joint analysis of groups of experiments, the Hartley F Max test was performed. All analyses were performed using the SAS statistical software, version 9.0 [37]. Figures were produced using the Sigma Plot software, version 10.

## 5. Conclusions

Our study is the first study in the Cerrado region to introduce the replacement of evapotranspiration in coffee plants. The results indicate that the responses of coffee plants to water deficits are based on the mechanisms of prevention and avoidance of drought: stomatal closure, reduction of transpiration and continuous water absorption. Physiological analysis showed that the avoidance of water loss by transpiration due to the control of stomatal opening and the rapid recovery of water by the leaf tissue seem to be the mechanisms that protect the plant during water stress and accelerate physiological recovery after the return of irrigation. However, yield data showed that, for these genotypes, the duration of stress and the 50% reduction in water availability limited the productivity of plants under water stress.

Irrigation supplementing rainfall is essential for coffee cultivation in the Cerrado, as rainfed cultivation is not favorable for the physiological and productive development of the crop. The use of the appropriate genotype for each region and each condition is important as genotypes showed both physiological and productive differences, and the choice will depend on the region in which it is to be cultivated and the desired productivity. Iapar 59 is a genotype recommended for production in the Cerrado due to the adapted physiology and high productivity, being superior to the other genotypes under full irrigation and water deficit management. E237 could potentially be used in genetic improvement programs as a source of genes for increased water deficit tolerance.

## Figures and Tables

**Figure 1 plants-11-02198-f001:**
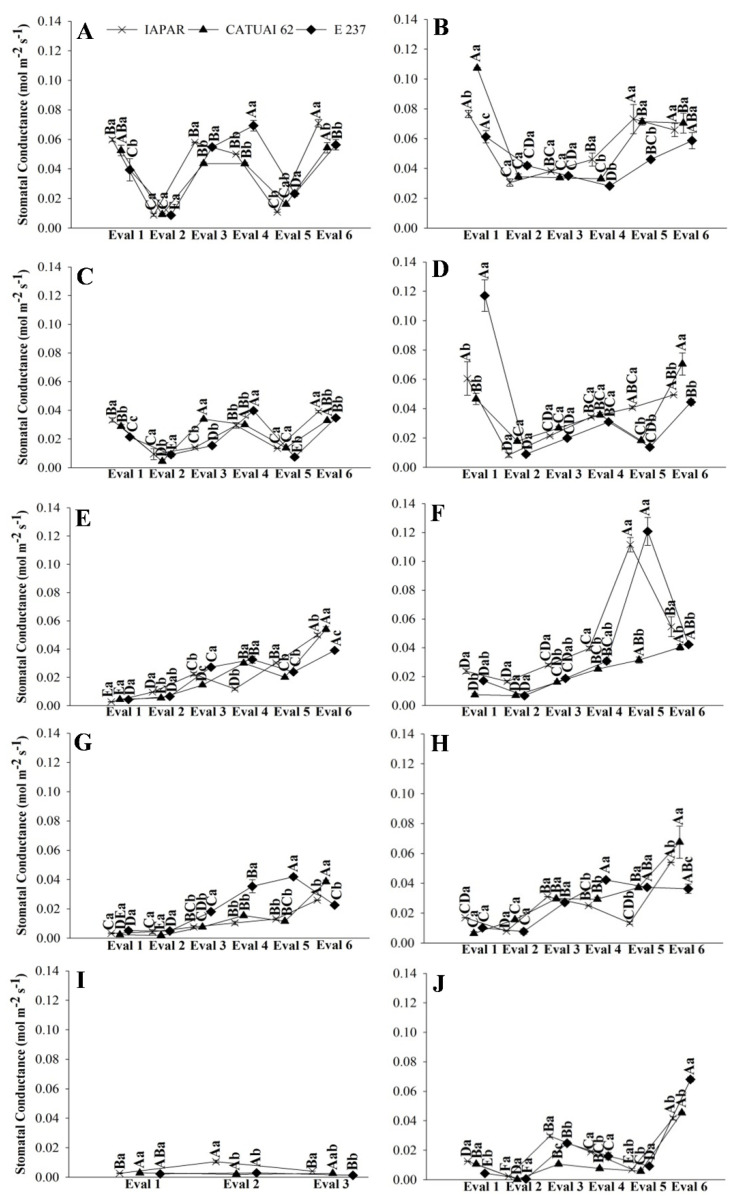
Stomatal conductance of coffee leaves under water stress (Eval 1) and after the return of irrigations (Eval 2 to 6), in 2019 and 2020, of three genotypes of arabica coffee (Iapar 59, Catuaí 62 and E237) subjected to five water regimes (FI 100 (**A**,**B**), FI 50 (**C**,**D**), WD 100 (**E**,**F**), WD 50 (**G**,**H**) and rainfed (**I**,**J**)). Capital letters compare the water regimes for each cultivar and lowercase letters compare the cultivars of each water regime. Means followed by the same letter do not differ according to the Tukey test at 5% probability. FI 100 and 50 (full irrigation with 100% and 50% replacement of evapotranspiration); WD 100 and 50 (water deficit with the suspension of irrigation from June to September, with replacement of 100% and 50% of evapotranspiration, respectively); rainfed (without irrigation).

**Figure 2 plants-11-02198-f002:**
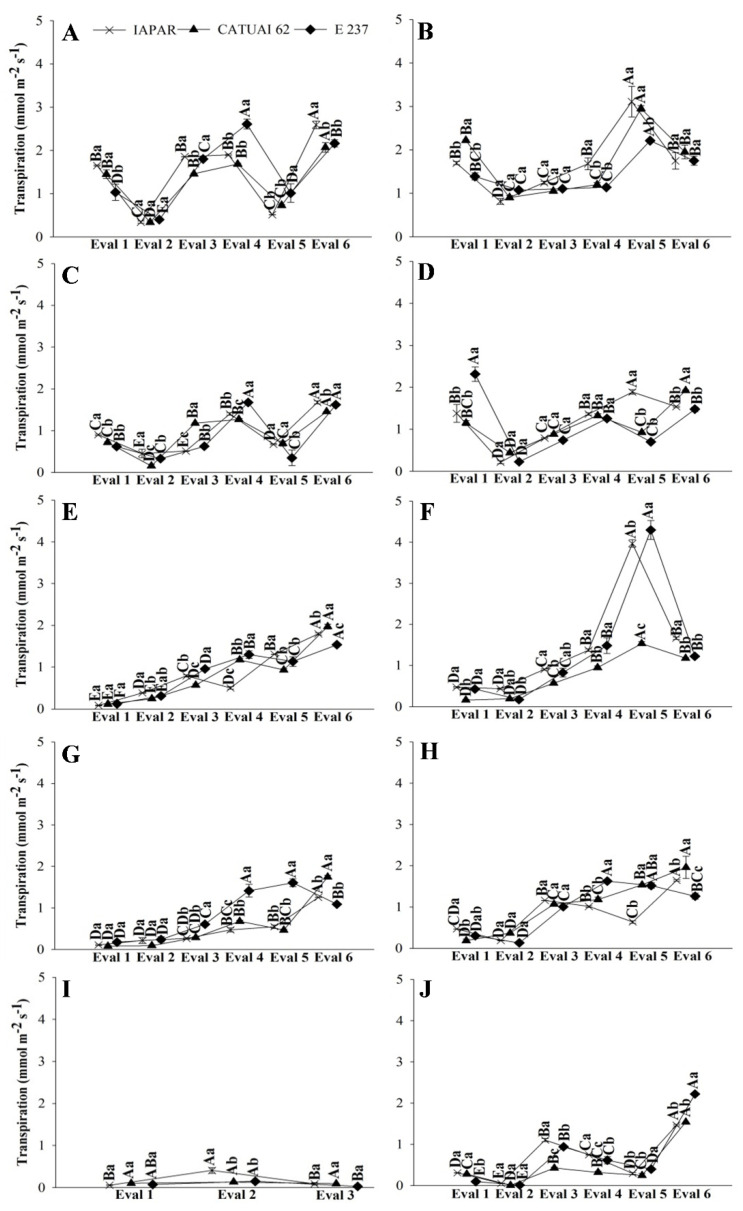
Transpiration of coffee leaves under water stress (Eval 1) and after the return of irrigation (Eval 2 to 6), in 2019 and 2020, of three genotypes of arabica coffee (Iapar 59, Catuaí 62 and E237) subjected to five water regimes (FI 100 (**A**,**B**), FI 50 (**C**,**D**), WD 100 (**E**,**F**), WD 50 (**G**,**H**) and rainfed (**I**,**J**)). Capital letters compare the water regimes for each cultivar and lowercase letters compare the cultivars of each water regime. Means followed by the same letter do not differ according to the Tukey test at 5% probability. FI 100 and 50 (full irrigation with 100% and 50% replacement of evapotranspiration); WD 100 and 50 (water deficit with the suspension of irrigation from June to September, with replacement of 100% and 50% of evapotranspiration, respectively); rainfed (without irrigation).

**Figure 3 plants-11-02198-f003:**
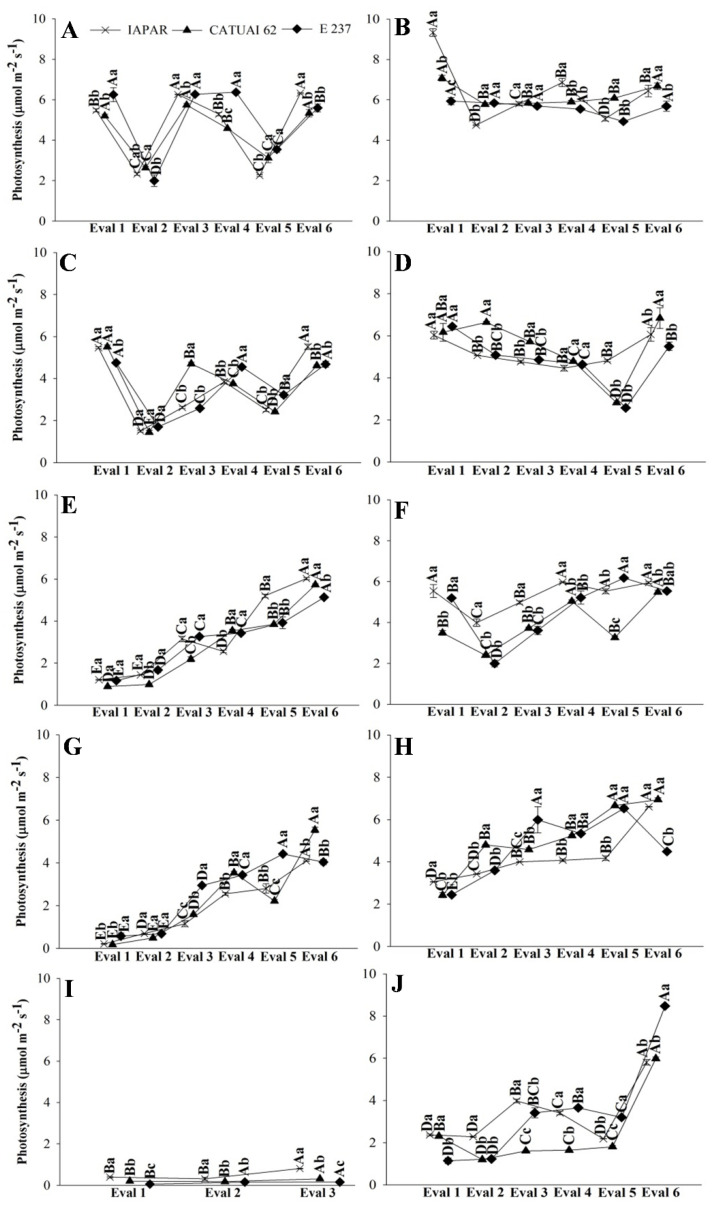
Photosynthetic rate of coffee leaves under water stress (Eval 1) and after the return of irrigation (Eval 2 to 6), in 2019 and 2020, of three genotypes of arabica coffee (Iapar 59, Catuaí 62 and E237) subjected to five water regimes (FI 100 (**A**,**B**), FI 50 (**C**,**D**), WD 100 (**E**,**F**), WD 50 (**G**,**H**) and rainfed (**I**,**J**)). Capital letters compare the water regimes for each cultivar and lowercase letters compare the cultivars of each water regime. Means followed by the same letter do not differ according to the Tukey test at 5% probability. FI 100 and 50 (full irrigation with 100% and 50% replacement of evapotranspiration); WD 100 and 50 (water deficit with the suspension of irrigation from June to September, with replacement of 100% and 50% of evapotranspiration, respectively); rainfed (without irrigation).

**Figure 4 plants-11-02198-f004:**
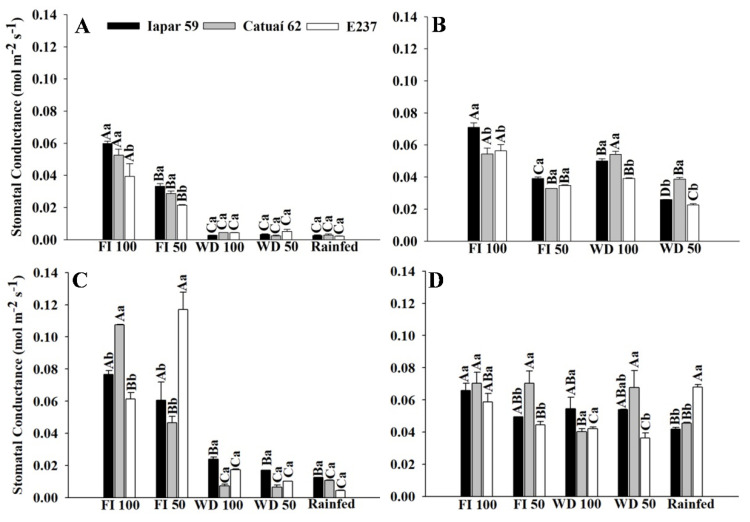
Stomatal conductance of coffee leaves under water stress (**A**,**C**) and after irrigation (**B**,**D**), in 2019 (**A**,**B**) and 2020 (**C**,**D**), of three arabica coffee genotypes (Iapar 59, Catuaí 62 and E237) under five water regimes (FI 100, FI 50, WD 100 WD 50 and rainfed). Capital letters compare the water regimes for each cultivar and lowercase letters compare the cultivars of each water regime. Means followed by the same letter do not differ according to the Tukey test at 5% probability. FI 100 and 50 (full irrigation with 100% and 50% replacement of evapotranspiration); WD 100 and 50 (water deficit with the suspension of irrigation from June to September, with replacement of 100% and 50% of evapotranspiration, respectively); rainfed (without irrigation).

**Figure 5 plants-11-02198-f005:**
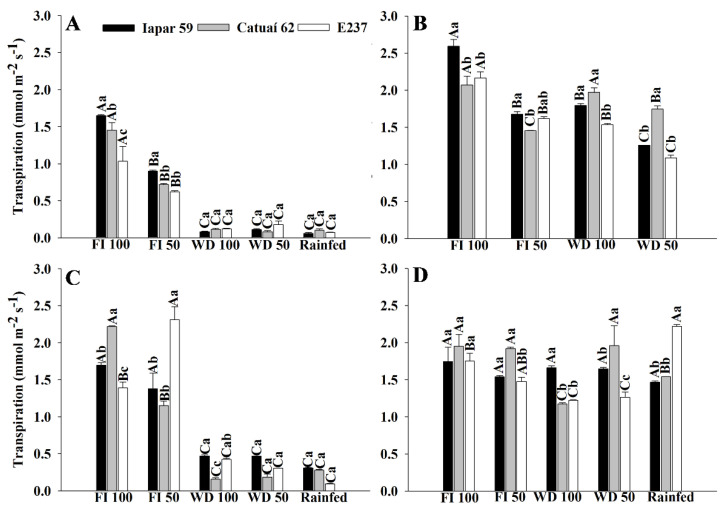
Transpiration of coffee leaves under water stress (**A**,**C**) and after irrigation (**B**,**D**), in 2019 (**A**,**B**) and 2020 (**C**,**D**), of three arabica coffee genotypes (Iapar 59, Catuaí 62 and E237) under five water regimes (FI 100, FI 50, WD 100, WD 50 and rainfed). Capital letters compare the water regimes for each cultivar and lowercase letters compare the cultivars of each water regime. Means followed by the same letter do not differ according to the Tukey test at 5% probability. FI 100 and 50 (full irrigation with 100% and 50% replacement of evapotranspiration); WD 100 and 50 (water deficit with the suspension of irrigation from June to September, with replacement of 100% and 50% of evapotranspiration, respectively); rainfed (without irrigation).

**Figure 6 plants-11-02198-f006:**
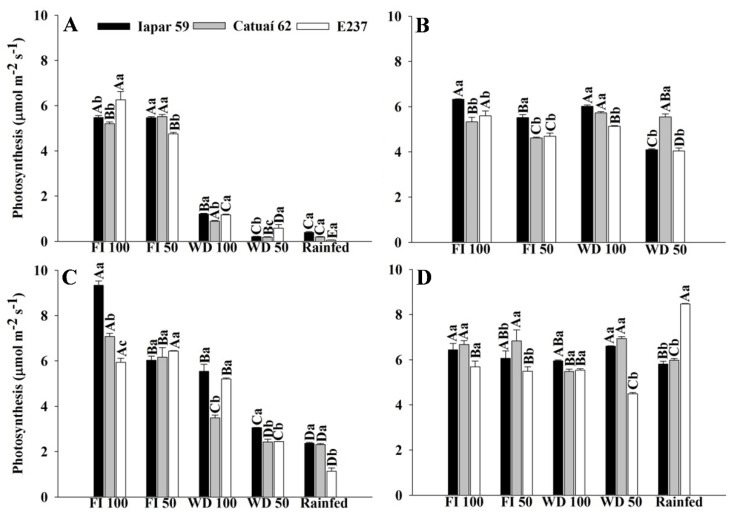
Photosynthetic rates of coffee leaves under water stress (**A**,**C)** and after irrigation (**B**,**D**), in 2019 (**A**,**B**) and 2020 (**C**,**D**), of three arabica coffee genotypes (Iapar 59, Catuaí 62 and E237) under five water regimes (FI 100, FI 50, WD 100, WD 50 and rainfed). Capital letters compare the water regimes for each cultivar and lowercase letters compare the cultivars of each water regime. Means followed by the same letter do not differ according to the Tukey test at 5% probability. FI 100 and 50 (full irrigation with 100% and 50% replacement of evapotranspiration); WD 100 and 50 (water deficit with the suspension of irrigation from June to September, with replacement of 100% and 50% of evapotranspiration, respectively); rainfed (without irrigation).

**Figure 7 plants-11-02198-f007:**
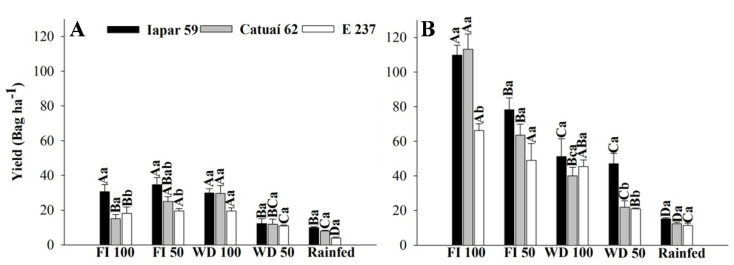
Productivity (bag ha^−1^) of three arabica coffee genotypes (E237, Iapar 59 and Catuaí 62) under five water regimes (FI 100, FI 50, WD 100, WD 50 and rainfed), in 2019 (**A**) and 2020 (**B**). Capital letters compare the water regimes for each cultivar and lowercase letters compare the cultivars of each water regime. Means followed by the same letter do not differ according to the Tukey test at 5% probability. FI 100 and 50 (full irrigation with 100% and 50% replacement of evapotranspiration); WD 100 and 50 (water deficit with the suspension of irrigation from June to September, with replacement of 100% and 50% of evapotranspiration, respectively); rainfed (without irrigation). Bag = 60 kg.

**Figure 8 plants-11-02198-f008:**
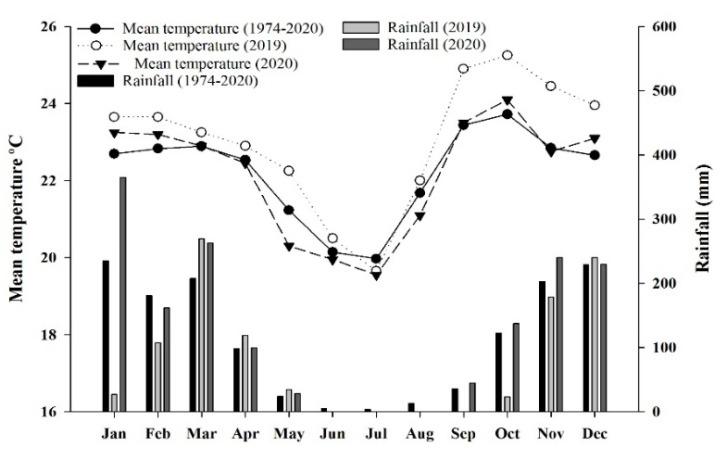
Climatic data of precipitation and mean temperature from 1974 to 2020 in the experimental area.

**Figure 9 plants-11-02198-f009:**
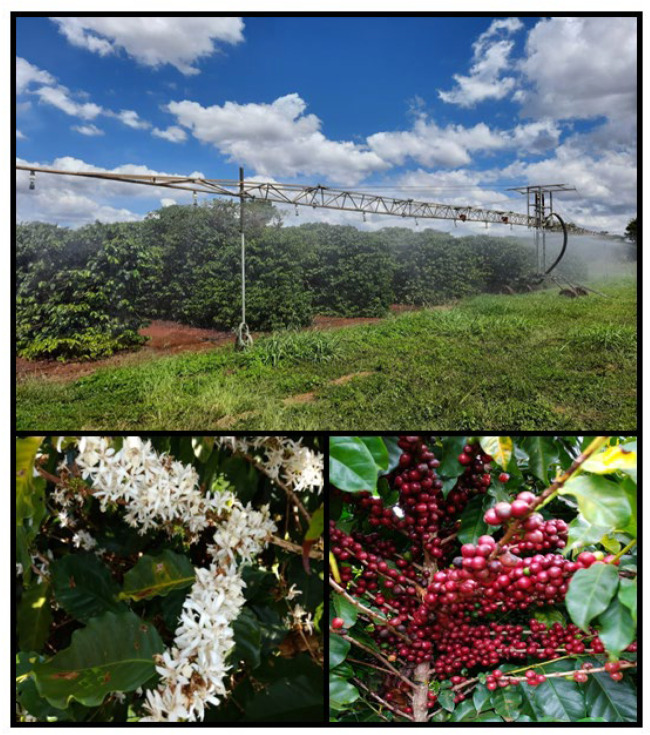
Irrigation system with controlled stress, uniform flowering and cherry grains.

**Table 1 plants-11-02198-t001:** Relative water content (RWC) in leaves of three coffee arabica genotypes (E237, Iapar 59 and Catuaí 62) under five water regimes (FI 100%, FI 50%, WD 100%, WD 50% and rainfed), during water stress in 2019 and 2020.

Water Regime	Genotype					
	E237	Iapar 59	Catuaí 62	E237	Iapar 59	Catuaí 62
	2019			2020		
FI 100	78.92 Aa	76.90 Aa	74.95 Aa	61.72 ABb	77.78 Aa	77.53 Aa
FI 50	73.07 Ba	73.75 Aa	73.24 Aa	73.94 Aa	71.84 Aa	72.97 Aa
WD 100	36.04 CDb	40.15 Bab	43.04 Ba	42.42 Cb	54.36 Ba	52.95 Ba
WD 50	31.83 Da	33.42 Ca	31.34 Ca	45.65 Ca	49.05 Ba	49.77 Ba
Rainfed	38.02 Ca	38.09B Ca	31.98 Cb	50.11B Ca	49.72 Ba	43.18 Ba

Capital letters compare the water regimes for each cultivar and lowercase letters compare the cultivars of each water regime. Means followed by the same letter do not differ according to the Tukey test at 5% probability. FI 100 and 50 (full irrigation with 100% and 50% replacement of evapotranspiration); WD 100 and 50 (water deficit with the suspension of irrigation from June to September, with replacement of 100% and 50 of evapotranspiration, respectively); rainfed (without irrigation).

**Table 2 plants-11-02198-t002:** Effect of water regimes (FI 100%, FI 50%, WD 100%, WD 50% and rainfed) in terms of relative water content (RWC) in leaves of coffee arabica genotypes after the return of irrigation, in 2019 and 2020.

Water Regime	RWC (%)	
	2019	2020
FI 100	70.82 b	78.43 a
FI 50	72.34 ab	78.54 a
WD 100	76.13 a	76.36 ab
WD 50	71.22 b	76.30 ab
Rainfed	32.95 c	74.35 b

Means followed by the same letter in each column do not differ according to the Tukey test at 5% probability. FI 100 and FI 50 (full irrigation with 100% and 50% replacement of evapotranspiration); WD 100 and 50 (water deficit with the suspension of irrigation from June to September, with replacement of 100% and 50 of evapotranspiration, respectively); rainfed (without irrigation).

**Table 3 plants-11-02198-t003:** Relative water content (RWC) in coffee arabica genotypes (E237, Iapar 59 and Catuaí 62), after the return of irrigation in 2019 and 2020.

Genotype	RWC (%)	
	2019	2020
E237	65.07 ab	80.07 a
Iapar 59	66.09 a	75.17 b
Catuaí 62	62.92 b	75.15 c

Means followed by the same letter in each column do not differ according to the Tukey test at 5% probability.

## Data Availability

All coffee genotypes used in this manuscript are released on the market and registered with the Ministry of Agriculture: https://sistemas.agricultura.gov.br/snpc/cultivarweb/cultivares_registradas.php (accessed on 15 July 2022).

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
