# Peer review of "Physiological Changes of Arabica Coffee under Different Intensities and Durations of Water Stress in the Brazilian Cerrado"

_plants, 2022, doi:10.3390/plants11172198_

Round 1

Reviewer 1 Report

General comments

I have read the manuscript (plants-1841848). Entitle: Physiological changes of Arabica coffee under different intensities and durations of water stress in the Brazilian Cerrado written by Patricia Carvalho da Silva et. al., for publication of life MDPI. In this study, the author investigates the evaluate the responses to different levels and duration of water deficit in arabica coffee genotypes. In this study, the author evaluated the gas exchange (photosynthesis, stomatal conductance, and transpiration), relative water content, and productivity.  In this study author mainly found that the plants under the FI 100 water regime showed better performance; however, after irrigation, coffee plants under WDs recovered their photosynthetic potential, showing similar results to FI100. Moreover, the author also found that phenotype Iapar 59 was the most productive genotype and E237 the lowest. Under rainfed conditions, the plants showed lower physiological and productive potentials, indicating the importance of irrigation in Coffea arabica in the Brazilian Cerrado.

The overall research is well conducted, and research is obvious application potential for the readers because this study mainly focuses on the coffee arabica genotype in different water stress levels and their resistance power and its cultivation expansion and breeding program based on the suitability for multiple environments in the modern era of the changing climate. In this sense, the manuscript is much valuable. However, I found some points, especially the flow of the text and lack of potential references, and lack of connection of story in different paragraphs, especially in the introduction and discussion sections. The author should provide enough examples and their interpretation of different traits of physiological and biochemical responses by the latest and appropriate references, some of which I mentioned below. Overall after I evaluate and request the author for this manuscript as a “MAJOR REVISION”

Major suggestions

1)Abstract: Abstract is cnot well written, its methodological part seems more focus rather than the results and its future insights. Author should mention a little bit more result parts by comparing the genotypes of coffee again the deficit conditions. Abstract is the main part of the manuscript therefore it should be perfect. It should be concise and short, and more informative but I think it’s not so well written and your abstract crosses the word limit under the journal rule please check this and correct t.

 2)  Introduction: The introduction is well starting with the background and the economic importance of coffee arabica and the change in production based on climate change and n earth’s water cycle which is much appreciated. However, the overall effect of drought needs to mention primarily, which I do not see in this introduction. This may be the appropriate reference to describe the negative effect of the drought as a reference. Entitle: Entitle “Response of drought stress in prunus sargentii and larix kaempferii ...https://doi.org/10.1016/j.foreco.2020.118099” Please mentioned that “drought reduced the morphological and physiological traits, reduce the leaf water potential and sap movement due to alternation of xylem anatomical features in the plants”. Then only write the background of Bletilla striata of medically important and other benefits.

3)     3) Hypothesis and objectives in the introduction: Author, should make more clearly present the research hypothesis and its objectives parts more clearly. The author should mention in the Ln 71-78 but it should focus on the hypothesis of the study which should be well connected with the objectives. The hypothesis should be very clear in the introduction sections because, without appropriate literature, questions, or hypotheses in the introduction section the entire text will be unclear. 

  Some other comments

4) Line no. 63-66: The author should Improve this line further by describing the plant water relation under drought by referring these references. Generally, the stomatal conductance (gs) plays a very important role in controlling the mechanism for the plant water status.  I request to author to refer to these articles 1) https://DOI:10.1016/j.scienta.2018.11.021 (2) doi: 10.3390/plants8070232. describe the controlled by the stomata opening and closing because stromal full or partial opening causes the turgid the plant part that also determines the plant water use efficiency”.

 5) Line no. 118:  The statistical part inside the figures and the x-axis and Y-axis are not much clear. I request to author to improve those letters (statistical analysis part) by increasing the letter font a little bigger. Accordingly, the author should improve this in other figures too.

6) Line no. 183:  The relative water content, is the author calculated relative water content (RWC) of their water regimes while comparing with the respective control?  if not please mention some text related to this especially in the methodology section because soil water content always does not represent truly the available water content for the plant.

7) Line no. 255-257:  The gas exchange parameters such as net photosynthesis, transpiration, and stomatal conductance are sensitive indicators under drought. https://doi.org/10.1016/j.scitotenv.2021.146466 refer and cite this article in line no 257 which is well described related to all related drought indicators under drought stress conditions to evaluate the different species/genotypes/cultivars adapted to environments with limited water availability.

8) Conclusion: The conclusion should be mentioned separately, and it should not be repetitive of the abstract or a summary of the results section. I would love to read striking points and take-home messages that will linger in the readers’ minds. What is the novelty, how does the study elucidate some questions in this field, and the contributions the paper may offer to the scientific community?

9) Line no. 475 (Reference): please double-check the citations, their style, spell check, and other grammatical errors. moreover, I request to the authors for revision throughout the manuscript according to the journal rules.

 Good Luck!

Author Response

Responses to the first reviewer:

 for the manuscript (ID: plants-1841848) “Physiological changes of Arabica coffee under different intensities and durations of water stress in the Brazilian Cerrado” written by Patricia Carvalho da Silva et al.

The popularity of coffee is well known. Therefore, any new scientific data regarding this culture always deserve attention. The manuscript presents new research data on the influence of water regime variations (water is one of the key factors for plant growth and development) on photosynthesis by using a portable gas exchange (photosynthesis) system LI-6400XT, relative water balance in leaves, and productivity of three Coffea arabica genotypes Iapar 59, Catuaí 62, and E237 with different tolerance to drought under field conditions in Cerrado areas. The aim of this research is also to identify mechanisms of drought tolerance and irrigation responsiveness for coffee cultivation in Cerrado areas. In my opinion, this manuscript is interesting in scientific and practical terms and might be recommended for publication after revision according to the following comments.

Reviewer: It is necessary to edit the text avoiding the repetition of material in sections “Results” and Discussion” and unsuccessful expressions, such as “The seedlings of the genotypes used in the studies were produced using seeds supplied by Embrapa Café (introductions from Ethiopia) and IAPAR, with materials already tested for drought tolerance under controlled conditions”? . (lines 369–371). Simply speaking “we used seeds”….

Authors: We corrected the text and in this version with track changes, the corrected text is between lines; 442 to 446.

Reviewer: It is desirable to unload the presentation of the material by removing figures 1–3, since the histograms contain enough information, and improve the captions to the histograms for its easier perception by the reader.

Authors: we justify the maintenance of the figures because they show us all the recuperation of each genotype after the water stress in the stressed treatments compared to the irrigated treatments (IF 100 and IF 50).

Reviewer: In the conclusions, clearly state the authors' ideas about the mechanisms of drought tolerance and irrigation responsiveness of coffea genotypes and practical recommendations for coffee cultivation in Cerrado areas based on the data obtained.

Authors: We included the conclusions.

Reviewer: It is advisable to add photographs of the investigated objects – trees or branches with flowers and fruits to decorate the manuscript for readers.

 Authors: We included some fotos showing the irrigation system, flowers of coffee tree and cherry grains (they are in figure 9).

Author Response

Responses to the second reviewer:

I have read the manuscript (plants-1841848). Entitle: Physiological changes of Arabica coffee under different intensities and durations of water stress in the Brazilian Cerrado written by Patricia Carvalho da Silva et. al., for publication of life MDPI. In this study, the author investigates the evaluate the responses to different levels and duration of water deficit in arabica coffee genotypes. In this study, the author evaluated the gas exchange (photosynthesis, stomatal conductance, and transpiration), relative water content, and productivity.  In this study author mainly found that the plants under the FI 100 water regime showed better performance; however, after irrigation, coffee plants under WDs recovered their photosynthetic potential, showing similar results to FI100. Moreover, the author also found that phenotype Iapar 59 was the most productive genotype and E237 the lowest. Under rainfed conditions, the plants showed lower physiological and productive potentials, indicating the importance of irrigation in Coffea arabica in the Brazilian Cerrado.

The overall research is well conducted, and research is obvious application potential for the readers because this study mainly focuses on the coffee arabica genotype in different water stress levels and their resistance power and its cultivation expansion and breeding program based on the suitability for multiple environments in the modern era of the changing climate. In this sense, the manuscript is much valuable.

However, I found some points, especially the flow of the text and lack of potential references, and lack of connection of story in different paragraphs, especially in the introduction and discussion sections.

The author should provide enough examples and their interpretation of different traits of physiological and biochemical responses by the latest and appropriate references, some of which I mentioned below. Overall after I evaluate and request the author for this manuscript as a “MAJOR REVISION”

Major suggestions

Reviewer: 1)Abstract: Abstract is not well written, its methodological part seems more focus rather than the results and its future insights. Author should mention a little bit more result parts by comparing the genotypes of coffee again the deficit conditions. Abstract is the main part of the manuscript therefore it should be perfect. It should be concise and short, and more informative but I think it’s not so well written and your abstract crosses the word limit under the journal rule please check this and correct t.

Authors: We corrected the abstract and included more results in the paper. (lines 32 to 39).

Reviewer: 2)  Introduction: The introduction is well starting with the background and the economic importance of coffee arabica and the change in production based on climate change and n earth’s water cycle which is much appreciated. 

However, the overall effect of drought needs to mention primarily, which I do not see in this introduction.

This may be the appropriate reference to describe the negative effect of the drought as a reference. Entitle: Entitle “Response of drought stress in prunus sargentii and larix kaempferii ...https://doi.org/10.1016/j.foreco.2020.118099”

Please mentioned that “drought reduced the morphological and physiological traits, reduce the leaf water potential and sap movement due to alternation of xylem anatomical features in the plants”. Then only write the background of Bletilla striata of medically important and other benefits.

Authors: We corrected and included Much more information as requested by the reviewer, and the alterations in the introduciton are between lines 63 to 64, 67 to 70, 76 to 101, 110 to 114.

Reviewer: 3) Hypothesis and objectives in the introduction:

 Author, should make more clearly present the research hypothesis and its objectives parts more clearly. The author should mention in the Ln 71-78 but it should focus on the hypothesis of the study which should be well connected with the objectives. The hypothesis should be very clear in the introduction sections because, without appropriate literature, questions, or hypotheses in the introduction section the entire text will be unclear. 

Authors: We corrected and the new text is between lines 116 to 121 in the new version

  Some other comments

Reviewer: 4) Line no. 63-66: The author should Improve this line further by describing the plant water relation under drought by referring these references. Generally, the stomatal conductance (gs) plays a very important role in controlling the mechanism for the plant water status.  I request to author to refer to these articles 1) https://DOI:10.1016/j.scienta.2018.11.021 (2) doi: 10.3390/plants8070232. describe the controlled by the stomata opening and closing because stromal full or partial opening causes the turgid the plant part that also determines the plant water use efficiency”.

c: We included Much more information and they are between lines 76 to 101.

Reviewer:  5) Line no. 118:  The statistical part inside the figures and the x-axis and Y-axis are not much clear. I request to author to improve those letters (statistical analysis part) by increasing the letter font a little bigger. Accordingly, the author should improve this in other figures too.

Authors: We improved the resolution of all figures of the paper, and increased the letters in the figures.

Reviewer:  6) Line no. 183:  The relative water content, is the author calculated relative water content (RWC) of their water regimes while comparing with the respective control?  if not please mention some text related to this especially in the methodology section because soil water content always does not represent truly the available water content for the plant.

Authors: We calculated the RWC for each treatment. Each RWC was calculated for each treatment.

Reviewer:  7) Line no. 255-257:  The gas exchange parameters such as net photosynthesis, transpiration, and stomatal conductance are sensitive indicators under drought. https://doi.org/10.1016/j.scitotenv.2021.146466 refer and cite this article in line no 257 which is well described related to all related drought indicators under drought stress conditions to evaluate the different species/genotypes/cultivars adapted to environments with limited water availability.

Authors: We included this article. And add a text also between lines 330 to 333 of the new version.

Reviewer:  8) Conclusion: The conclusion should be mentioned separately, and it should not be repetitive of the abstract or a summary of the results section. I would love to read striking points and take-home messages that will linger in the readers’ minds. What is the novelty, how does the study elucidate some questions in this field, and the contributions the paper may offer to the scientific community?

Authors: We included the conclusion, and they are between lines 520 to 538

Reviewer:  9) Line no. 475 (Reference): please double-check the citations, their style, spell check, and other grammatical errors. moreover, I request to the authors for revision throughout the manuscript according to the journal rules.

Authors: We checked and reviewed.

Round 2

Reviewer 1 Report

Dear Author

I have read the revised manuscript (plants -1841848). Titled: Physiological changes of Arabica coffee under different intensities and durations of water stress in the Brazilian Cerrado for publication in plants MDPI. This is the second submission made by the author. The author addressed all the questions and suggestions that I raised the issue in the review of the original manuscript. I satisfy the author’s revisions throughout the paper. Author well addresses the abstract issues. Especially author improved the introduction and discussion section very well inflow. Now, this manuscript improved the flow of writing, which was comparatively shallow in the original version but in this revised copy author addressed all the quarries and suggestions very well. Before accepting this manuscript if there is anything needed to be revised by the author, especially English grammar, or spell check, I request this manuscript is currently in “Minor Revision” and any grammatical error author may improve in this stage. Thank you.